# Disease Resistant Citrus Breeding Using Newly Developed High Resolution Melting and CAPS Protocols for Alternaria Brown Spot Marker Assisted Selection

**Carmen Arlotta** [1,†]**, Angelo Ciacciulli** [1,†]**, Maria Concetta Strano** [1]**, Valeria Cafaro** [1]**,
Fabrizio Salonia** [1,2]**, Paola Caruso** [1]**, Concetta Licciardello** [1]**, Giuseppe Russo** [1]**,
Malcolm Wesley Smith** [3]**, Jose Cuenca** [4]**, Pablo Aleza** [4] **and Marco Caruso** [1,*]

[1] CREA-Research Centre for Olive, Fruit and Citrus crops—Corso Savoia 190, 95024 Acireale (CT), Italy;
carmen.arlotta@crea.gov.it (C.A.); angelo.ciacciulli@crea.gov.it (A.C.);
mariaconcetta.strano@crea.gov.it (M.C.S.); valeriacafaro96@gmail.com (V.C.); fabrizio.salonia@unict.it (F.S.);
paola.caruso@crea.gov.it (P.C.); concetta.licciardello@crea.gov.it (C.L.); giuseppe.russo@crea.gov.it (G.R.)

[2] Department of Agriculture, Food and Environment (Di3A), University of Catania—Via Valdisavoia 5,
95123 Catania, Italy

[3] Department of Agriculture and Fisheries, 49 Ashfield Road, Bundaberg, QLD 4670, Australia;
Malcolm.Smith@daf.qld.gov.au

[4] Centro de Citricultura y Producción Vegetal, Instituto Valenciano de Investigaciones Agrarias (IVIA),
Moncada, 46113 Valencia, Spain; cuenca_josiba@gva.es (J.C.); aleza_pab@gva.es (P.A.)

[*] Correspondence: marco.caruso@crea.gov.it; Tel.: +39-0957653119

[†] These authors contributed equally to this work.

**Abstract:** *Alternaria alternata* is a fungus that causes a serious disease in susceptible genotypes of citrus, particularly in mandarins. The *Alternaria citri* toxin (ACT) produced by the pathogen induces necrotic lesions on young leaves and fruits, defoliation and fruit drop. Here, we describe two methods of marker-assisted selection (MAS) that could be used for the early identification of Alternaria brown spot (ABS)-resistant mandarin hybrids. The first method is based on a nested PCR coupled to high resolution melting (HRM) analysis at the SNP08 locus, which is located at 0.4 cM from the ABS resistance locus, and was previously indicated as the most suitable for the selection of ABS-resistant hybrids. The method was validated on 41 mandarin hybrids of the CREA germplasm collection, and on 862 progenies generated from five crosses involving different susceptible parents. Four out of five populations showed Mendelian segregation at the analyzed locus, while a population involving Murcott tangor as male parent showed distorted segregation toward the susceptible hybrids. The second method is based on a cleaved amplified polymorphic sequences (CAPS) marker that was developed using the same primers as the nested PCR at the SNP08 locus, coupled with *Bcc*I restriction enzyme digestion. To verify the reliability of the two genotyping methods, in vitro leaf phenotyping was carried out by inoculating *A. alternata* spores onto young leaves of 101 hybrids, randomly chosen among the susceptible and resistant progenies. The phenotyping confirmed the SNP08 genotyping results, so the proposed method of selection based on HRM or CAPS genotyping could be routinely used as an alternative to KBioscience competitive allele specific polymerase chain reaction (KASPar) single nucleotide polymorphism (SNP) genotyping system to improve citrus breeding programs. While the study confirmed that the SNP08 marker is a reliable tool for MAS of new citrus hybrids with different genetic backgrounds, it also identified a small group of genotypes where the resistance mechanism requires further investigation.

**Keywords:** *Alternaria alternata*; genotyping; mandarins; single nucleotide polymorphisms

---

## 1. Introduction

Genetic progress in citrus breeding programs is hampered by the complex reproductive biology, the high level of heterozygosity, large plant size, and the long juvenile period. In addition, there is a very limited number of markers that can be used for marker assisted selection (MAS) to facilitate the early selection of new varieties for specific, desirable traits [1]. However, progress has been made in the last few years with the identification of candidate genes or loci associated with important pomological and disease resistance traits, through conventional mapping and Genome-Wide Association Study (GWAS) approaches [2]. Markers for assisted selections are now available for anthocyanin pigmentation [3], nucellar embryony [4], Alternaria brown spot (ABS) resistance [5] and CTV resistance [6].

ABS is caused by the fungus *Alternaria alternata* (Fr.) Keissl, whose germinated spores produce an *Alternaria citri* toxin (ACT) that kills the host cells, so that the pathogen can utilize the released nutrients. The pathogen causes brown-to-black necrotic lesions surrounded by a yellow halo in twigs, young leaves, and fruits [7–9], inducing defoliation and fruit drop.

Symptoms under field conditions are particular severe in some mandarins and grapefruit (*C. paradisi* Macf.) [10], while artificial inoculum may cause symptoms in additional accessions such as sweet orange [11,12]. Economically important mandarin cultivars, such as 'Emperor', 'Fortune', 'Nova' and 'Murcott', are susceptible to ABS [13]. These cultivars are grown for the fresh fruit market where unsightly lesions down-grade value and cause an important economic loss [14]. 'Emperor' was grown commercially as a fresh fruit variety in Australia until it was decimated by ABS [15], and the ongoing cost of control measures and fruit losses in 'Murcott' has been estimated at more than USD $3000 per hectare [16]. Bassimba et al. [17] reported a drastic reduction in the production of 'Fortune' in the Valencian community (Spain), from 161.000 t to 21.000 t, from the first ABS outbreak in 1998 until 2012. More recently, de-Miguel et al. [18] reported the almost complete disappearance of 'Fortune' in Spain due to its ABS susceptibility.

Performing MAS for ABS resistance might help to sustain modern citriculture facilitating the generation of new resistant citrus cultivars. The control of ABS usually requires many fungicide treatments (4 to 16 per annum, depending on the genotype and environment) and genetic resistance remains the best long-term solution. Therefore, the early stage selection of resistant hybrids through molecular markers is an affordable and efficient strategy to obtain resistant cultivars and improve the genetic structure of breeding populations [2].

To map the ABS resistance locus, a strategy based on bulked segregant analysis and single nucleotide polymorphism (SNP) genotyping has been used [19]. In a later study, fine mapping identified a set of SNPs associated with the resistance locus and analyzed by KBioscience® services, using the KASPar technique [20], which is one of the most reliable and efficient methods for high throughput SNP genotyping, based on fluorescence resonance energy transfer (FRET) and allele-specific oligo extension for signal generation [21]. KASPar utilizes highly specific 5'–3' exonuclease-deleted Taq DNA polymerase in combination with two allele-specific, tailed forward primers and one common reverse primer [20].

Among the SNP markers associated to the ABS resistance locus, SNP08 mapped at 0.4 cM (chromosome 3, position: 25862085) from the ABS resistance locus was chosen as the most suitable for the early selection of resistant hybrids. SNP08 is diallelic (G/T), with the G allele phased with susceptibility, such that GT and GG genotypes are susceptible, while only TT genotypes are resistant to ABS [5].

KASPar assays are simple methods to determine SNP genotypes, but require specific equipment or a proprietary service. Besides KASPar, several other methods are reported for SNP analysis, such as high resolution melting (HRM) and cleaved amplified polymorphic sequences (CAPS) markers. HRM analysis is an efficient and low-cost method that can be applied to detect genetic variations, including insertions or deletions (INDELs), microsatellites (simple sequence repeats—SSRs) and SNPs in PCR amplicons [22–24]. The technique uses a highly accurate optical detection system connected to a real-time PCR machine, and measures the change in fluorescence accompanied by the fusion of

double-stranded DNA using an intercalating dye of saturated DNA [25]. A single nucleotide difference causes a detectable change in the melting curve, and makes it possible to distinguish allelic differences between amplicons. Heterozygous genotypes are identified by a change in the shape of the melting curve, and different homozygotes are distinguished by a change in the melting temperature [26]. HRM has been used in different citrus species and varieties to analyze SSR [24,27] and SNP [25,27] polymorphisms. HRM melting curves are also helpful in discriminating zygotic from nucellar seedlings in rootstock breeding when highly polyembryonic female parents are used [28]. The HRM approach has a high sensitivity and can be applied to distinguishing genotypes when more than one SNP occurs within a single pair of PCR primers [22,29].

CAPS markers are developed from PCR products that undergo digestion with specific restriction enzymes followed by agarose gel separation. Based on the choice of the restriction enzyme, CAPS could be used for SNP genotyping if the difference among individuals is due to a base substitution in the amplified fragment. CAPS markers are codominant and require cheap laboratory equipment, but they are less adaptable to high throughput genotyping compared to other techniques [30]. In citrus, CAPS markers have been efficiently used for genetic diversity studies [31,32], parentage analysis [33], mapping [34], and intellectual property protection [35].

The present study reports the development of two complementary methods based on HRM and CAPS for the analysis of SNP08 in citrus, with validation on a large number of mandarin cultivars and progenies having different genetic backgrounds. These new protocols can be used as an alternative to KASPar genotyping at the ABS resistance locus using conventional laboratory equipment, such as real time PCR and common PCR, followed by agarose gel electrophoresis.

## 2. Materials and Methods

### 2.1. Plant Material and Recovery of Hybrids

Young leaves of 41 citrus accessions and of 862 progenies generated from five crosses involving parents susceptible to ABS were examined (Table 1). The 41 accessions belong to the CREA citrus germplasm of Acireale (Catania, Italy; 37°36′31″ N, 15°09′56″ E) and Palazzelli (Lentini, Siracusa, Italy; 37°20′22″ N, 14°53′31′′ E), and included mostly mandarin hybrids (Table 1). The hybrid populations involving an ABS susceptible parent were generated in 2018 and 2019 in the framework of the CREA breeding program. The evaluated hybrids were: ISA (resistant) × Seedless Kishu (susceptible); Foma107 (resistant) × Seedless Kishu; Cami (resistant) × Seedless Kishu; Rubino (resistant) × Murcott (susceptible); and Fortune (susceptible) × OTA9 (resistant). Fruits of the five crosses were collected in autumn 2018 and 2019 and seeds were extracted. Seed teguments were removed to facilitate germination, and the embryos were sown in Jiffy pots. Resulting plants were maintained in a climatic chamber with 16-h photoperiod at 25 °C, where young leaflets were sampled for DNA extraction when plants were about 2-months-old, and then again for spore inoculation when they were 3–5-months-old.

**Table 1.** Germplasm accessions used in the study, from CREA arboreta of Acireale and Palazzelli.

| Accession | Description | References |
|---|---|---|
| 50-15A-6 | 'Clementine' mandarin × 'Avana' mandarin/Unreleased parent of the CREA breeding program | [36] |
| Afourer | Possibly 'Murcott' tangor × 'Mandalina' mandarin | [37,38] |
| Avana (Willowleaf) | Old mandarin selection | [39] |
| Bower | 'Clementine' mandarin × 'Orlando' tangelo | [40] |
| Cami | '50-15A-6' mandarin × 'Mapo' tangelo | [36] |
| Carvalhais | Mandarin of unknown parentage | [39] |
| Clemapo | 'Clementine' mandarin × 'Mapo' tangelo/early-maturing hybrid released in the 1990's by a private breeding program | - |
| Daisy | 'Fortune' mandarin × 'Fremont' mandarin | [41] |
| Dancy | Seedling of Moragne tangierine | [39] |

**Table 1.** *Cont.*

| Accession | Description | References |
|---|---|---|
| Doppio Sanguigno | Sweet orange clonal selection | [42] |
| Ellendale | Tangor of unknown parentage | [39] |
| Emperor | 'Ponkan' mandarin selection | [38,39] |
| Encore | 'King' mandarin × 'Willowleaf' mandarin | [39] |
| Fairchild | 'Clementine' mandarin × 'Orlando' tangelo | [43] |
| Fallglo | 'Bower' mandarin ×' Temple' tangor | [44] |
| Foma107 | 'Fortune' mandarin × 'Mapo' tangelo/Unreleased parent of the CREA breeding program | - |
| Fortune | 'Clementine' mandarin × 'Orlando' tangelo | [45] |
| Fremont | 'Clementine' mandarin × 'Ponkan' mandarin | [39] |
| ISA | Old clementine clone of Italian origin | - |
| Kara | 'King' mandarin × 'Owari' satsuma | [39] |
| King | Natural tangor of Asian origin | [39] |
| Kinnow | 'King' mandarin × 'Willowleaf' mandarin | [39] |
| Malvasio | Mandarin of unknown parentage | [39] |
| Mapo | 'Avana' mandarin × 'Duncan' grapefruit | [46] |
| Michal | 'Clementine' mandarin × 'Dancy' mandarin | [38] |
| Minneola | 'Duncan' grapefruit × 'Dancy' mandarin | [39] |
| Murcott | Tangor of unknown parentage | [39] |
| Nova | 'Clementine' mandarin × 'Orlando' tangelo | [47] |
| Okitsu | Satsuma clonal selection | [39] |
| Ortanique | Tangor of unknown parentage | [39] |
| OTA9 | 'Oroval' clementine × 'Moro' sweet orange | [48] |
| Page | 'Minneola' tangelo × 'Clementine' mandarin | [39] |
| Palazzelli | 'Clementine' mandarin × 'King' mandarin | [49] |
| Ponkan | Old mandarin selection | [39] |
| Primosole | 'Miho' satsuma × 'Carvalhais' mandarin | [50] |
| Rubino | Late clementine clone | [51] |
| Seedless Kishu | Seedless mutation of 'Kishu' mandarin | [52] |
| Simeto | 'Miho' satsuma × 'Avana' mandarin | [50] |
| Star Ruby | Grapefruit clonal selection | [53] |
| Sunburst | 'Robinson' mandarin × 'Osceola' mandarin | [54] |
| Wilking | 'King' tangor × 'Willowleaf ' mandarin | [39] |

## 2.2. DNA Isolation

Genomic DNA was extracted from young leaves, following the method described in Caruso et al. [28] with minor modifications. Briefly, the isolation of DNA consisted of an initial homogenization of approximately 20 mg leaf samples in 300 μL CTAB extraction buffer (2% CTAB, 100 mM Tris–HCl, pH 8.0, 20 mM EDTA, pH 8.0, 1.4 M NaCl, and 0.1% 2-mercaptoethanol) in a 2 mL tube, using a Tissue Lyser (Qiagen, Valencia, CA, USA) and stainless steel beads, followed by incubation at 65 °C for 1 h. One hundred μL of chloroform was added to each tube, and the tube was vortexed and centrifuged for 10 min at about 10,000 g. DNA was precipitated by mixing 200 μL of supernatant with 500 μL 100% ethanol. The pellet was washed with 70% ethanol, dried to remove alcohol, and dissolved in water. DNA concentration was estimated by measuring UV absorption at 260 nm and 280 nm to assess DNA purity using a Nanodrop1000 spectrophotometer (Thermo Scientific, Wilmington, DE, USA).

## 2.3. Sequencing of the SNP08 Locus

Sanger sequencing was performed to create a reference for the interpretation of the HRM and CAPS genotyping. A subset of 10 mandarins, chosen among the susceptible and resistant genotypes (Avana, 50-15A-6, King, Fremont, ISA, Fortune, OTA9, Murcott, Rubino, and Minneola) were sequenced using the primers SNP08_nested PCR forward and SNP08_nested PCR reverse (Table 2). Moreover, due to the presence of more SNPs in the sequenced locus, two resistant and two susceptible hybrids from each population were sequenced with the same primers to reconstruct the haplotypes. GeneStudio™ Pro software version 2.2 (GeneStudio Inc., Suwanee, GA, USA) was used for contig assembly and SNPs identification.

**Table 2.** Oligo primers used for SNP08 selection.

| Primer Name | Primer Sequence (5'-3') | Ta (°C) | Product Size (bp) | Position in Chromosome 3 of the Clementine Genome |
| --- | --- | --- | --- | --- |
| SNP08_nested PCR | F: AACTCACAAACATGTCTTCAACAA<br>R: ATTGTCAATTGGTGGGAAGC | 57 | 572 | 25861812 … 25861844<br>25862364 … 25862383 |
| SNP08_HRM | F: AGCGAATAAATTTGATGCTGAGC<br>R: TCTTGTAGGACTAAAATTTCACTTGGA | 60 | 110 | 25862022 … 25862044<br>25862105 … 25862131 |

### 2.4. Bioinformatics Analysis

The detection of an allele-specific restriction site was done by the Benchling tool (https://www.benchling.com/) and a virtual digestion of both alleles was performed. The sequences of the amplicon and the flanking regions were blasted by BLAST+ [55], with the default setting on citrusgenomedb server (https://www.citrusgenomedb.org/) using the genomes of *C. maxima* and *C. reticulata* from HZAU [4] databases. The resulting duplications downloaded, in gff3 format, were decorated with the SNP08 position and plotted using the annotationsketch web tool (http://genometools.org/cgi-bin/annotationsketch_demo.cgi).

### 2.5. Nested-PCR

Forty-one mandarin accessions and 862 progenies of five populations generated from ABS susceptible parents were screened at the SNP08 locus using nested PCR coupled to HRM analysis. Primers for nested PCR (Table 2) were designed in the regions flanking the SNP08 identified by Cuenca [5] on chromosome 3 of the Clementine genome (position 25862085). PCR assays were performed in a reaction mixture of 10 μL, including: 2 μL of genomic DNA at variable concentration (i.e., no dilutions were made after DNA isolations), 1× DreamTaq Buffer (Thermo Fisher Scientific, Waltham, MA, USA), 0.16 mM dNTP, 0.16 μM forward and reverse primers and 0.5 U DreamTaq DNA polymerase (Thermo Fisher Scientific, Waltham, MA, USA). PCR was performed in a thermocycler with the following cycling program: an initial denaturation step of 94 °C for 3 min, followed by 35 cycles of 94 °C for 45 s, an optimal annealing temperature of 57 °C for 45 s, 72 °C for 1 min and a final extension at 72 °C for 15 min.

### 2.6. HRM Genotyping

Forward and reverse primers for HRM were designed to amplify a 110 bp amplicon flanking SNP08, to discriminate genotypes resistant and susceptible to ABS (Table 2). The reaction mixture of 15 μL contained 4 μL PCR product diluted 1:100, 1× DreamTaq Buffer, 0.16 mM dNTP, 0.16 μM forward and reverse primers, 1.6 μM Syto®9 (Invitrogen, Thermo Fisher Scientific, Waltham, MA, USA), and 0.75 U DreamTaq DNA polymerase.

HRM genotyping was performed on a StepOnePlus™ Real-time PCR system (Applied biosystems, Thermo Fisher Scientific, Foster city, CA, USA) and analyzed with the High Resolution Melt software version 3.0.1 (Applied biosystems, Thermo Fisher Scientific, Foster City, CA, USA; https://www.thermofisher.com/order/catalog/product/4461357#/4461357). The conditions were as follows: holding stage at 95 °C for 10 min; cycling stage 40 cycles of 95 °C for 15 s and 60 °C for 1 min; melt curve stage at 95 °C for 15 s; 60 °C for 1 min and 95 °C for 15 s. HRM analysis was performed at a ramp rate of 0.3% increments every cycle until 95 °C.

### 2.7. Development of Cleaved Amplified Polymorphic Sequences (CAPS) Marker

Restriction enzyme digestions of PCR amplicons of the nested PCR were performed in a final volume of 15 μL, which consisted of 6 μL PCR product, 4 U *Bcc*I restriction enzyme (New England Biolabs Inc., Ipswich, MA, USA), 1× CutSmart restriction enzyme buffer (New England Biolabs, Inc., Ipswich, MA, USA) and sterile water until final volume. The enzyme recognizes the region of five nucleotides "GATGG" in the amplicon, which corresponds to the position of the SNP08 in the presence

of G. The digestion time was 1 h at 37 °C in a water bath, and subsequent enzymatic inactivation for 20 min at 65 °C. The digestions were checked by electrophoresis on agarose gel at 1.2%, and SYBR safe (Invitrogen, Thermo Fisher Scientific, Waltham, MA, USA) was used as intercalant.

### 2.8. KASPar SNP Genotyping

One hundred and forty-eight hybrids from the Rubino × Murcott population were analysed for SNP08 developed by Cuenca et al. [5] using the KASPar technique, and the results were subsequently used as a reference for comparison with HRM genotyping. A detailed explanation of specific conditions and reagents can be found in Cuenca et al. [5,19].

### 2.9. Statistical Analysis

Segregation of the SNP08 locus was analyzed in each population using the chi-square test, assuming an expected allelic segregation ratio of 1:1 ($p < 0.05$).

### 2.10. Isolation and Production of A. alternata Inoculum

Symptomatic leaves of Nova mandarin were obtained from an orchard in Giarre (CT, N. 37.74338°– E. 15.18647°), and taken to the CREA laboratory. For the isolation of the fungus, small pieces of the foliar tissue with symptoms of the disease were first dipped in a solution of 1% sodium hypochlorite in water (1:3, *v/v*) for 1 min, then rinsed in sterile distilled water and placed in Petri dishes containing potato dextrose agar (PDA). Petri dishes were incubated at 25 ± 2 °C in darkness. After seven days, mycelium transfers were performed to obtain pure colonies of *A. alternata*. Pathogenicity of the fungal colonies was proven through inoculation on young Murcott tangor leaves. Pure colonies were stored in test tubes containing PDA medium at 4 °C.

To produce conidial suspension for inoculating leaves, 10 mL of sterile water was added to the surface of pure cultures; the surface was gently scraped with a sterile glass rod to remove the mycelium. Then, the suspension was filtered through two layers of sterile gauze; the spores were quantified in a Bürker counting chamber and concentration was adjusted to $10^5$ conidia·mL$^{-1}$.

### 2.11. Inoculation and Evaluation of Disease Severity in Detached Leaves

A total of 101 hybrids of Fortune × OTA9, Rubino × Murcott, ISA × Seedless Kishu, Foma107 × Seedless Kishu, and Cami × Seedless Kishu were evaluated for susceptibility to ABS. Three symptomless young leaves (4–6 cm in length) were collected per hybrid from 3–5 month-old plants grown in a climatic chamber. Leaves were dipped in a solution of 1% sodium hypochlorite in water (1:3, *v/v*) for 1 min, rinsed twice in sterile distilled water, then dried with sterile paper and placed in Petri dishes containing a layer of filter paper moistened with sterile distilled water (to create a microenvironment of high humidity). Leaves were inoculated by spraying the conidial suspension of the *A. alternata* isolate NF104 (approximately 1 mL) with a portable hand spray, over the lower surface of each leaf. Negative controls were inoculated by spraying sterile distilled water on leaves, while a positive control was obtained by spraying the inoculum onto an agar plate. The inoculated leaves were placed in a humid chamber and incubated at 25 ± 2 °C according to De Souza et al. [56], with modifications. The first evaluation was performed 48 h after the inoculation, and then for two subsequent days. A genotype was considered susceptible when leaf lesions were observed, and resistant if symptoms were absent.

## 3. Results and Discussion

### 3.1. Sequencing and Analysis of the Locus Flanking SNP08 and Development of the HRM Protocol

Before performing HRM genotyping, the SNP08 locus of some of the mandarin varieties and hybrids was sequenced, to be used as a golden standard for the interpretation of the HRM results. The sequencing of 10 varieties from the CREA collection (Figure 1) revealed that SNP08 was homozygous G/G in Minneola and T/T in Avana, Fremont, 50-15A-6, King, ISA (Clementine), Rubino, and OTA9,

and that it was heterozygous G/T in Fortune and Murcott. The current result is consistent with previous findings [5,19], except that Fremont was T/T, while it was G/G in the KASPar genotyping data reported by Cuenca et al. [5]. Furthermore, an additional diallelic SNP (C/T) located 18 bp downstream of SNP08 locus, was detected (SNP08+18bp). The SNP08+18bp was homozygous (C/C) in ISA, OTA9 and Minneola, and heterozygous (C/T) in Avana, 50-15A-6, King, and Fremont, showing different allelic frequencies (i.e., a higher C peak and a lower T in the electropherograms; Figure 1). The heterozygosity in Murcott showed the same allele frequency of C and T.

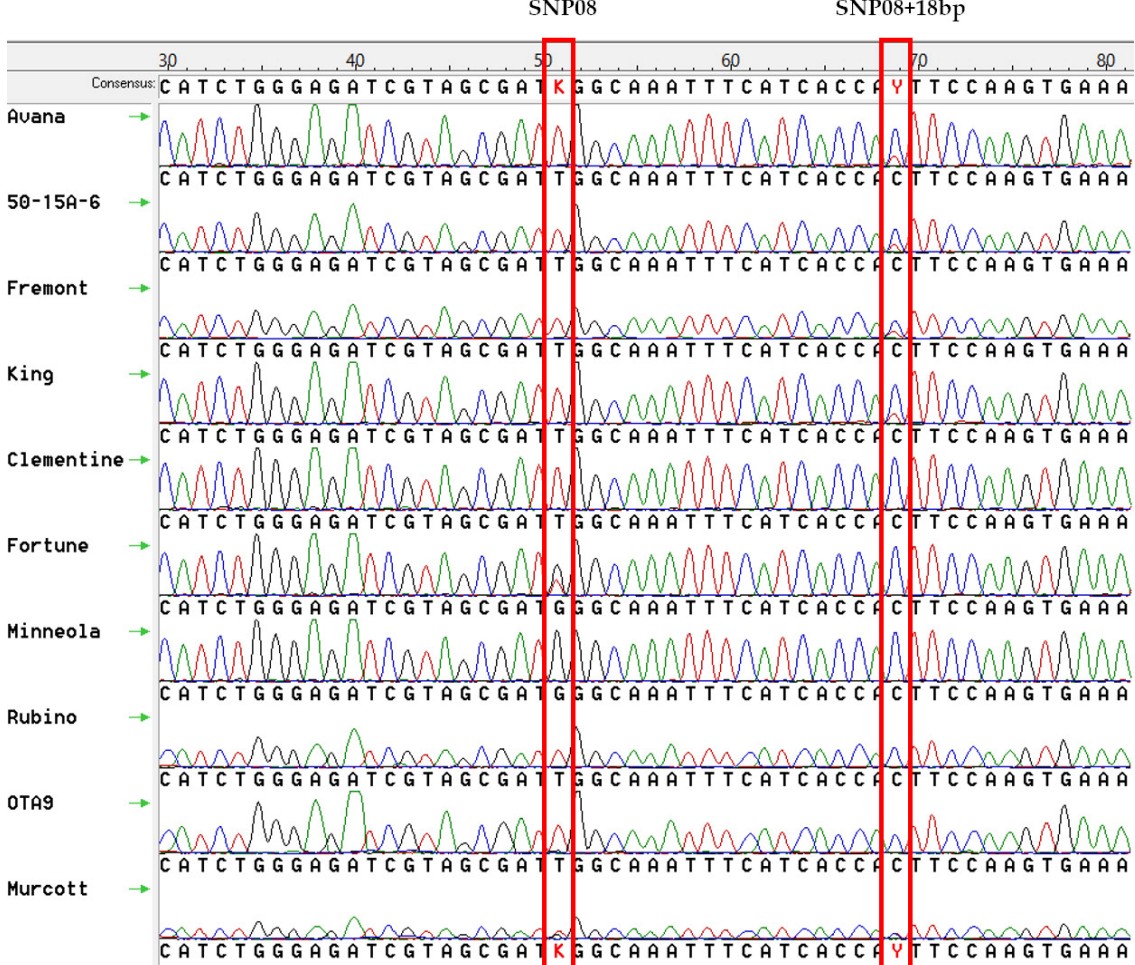

**Figure 1.** Alignment of the sequences flanking the SNP08 and additional SNP08+18bp in Avana, 50-15A-6, King, Fremont, ISA (Clementine), Rubino, and OTA9 (resistant accessions) and in Fortune, Minneola and Murcott (susceptible accessions).

The heterozygosity at the SNP08+18bp with different variant allele frequencies (VAFs) (i.e., different percentages of sequence reads with C and T) is also clear in the resequencing data retrieved from the citrus genome database (www.citrusgenomedb.org), where re-sequenced genomes of some citrus varieties have been aligned over the *C. clementina* genome (Figure 2). The SNP08+18bp was heterozygous (C/T), with different VAFs in Avana, W. Murcott, Ponkan, Seedless Kishu and sweet orange, and homozygous (C/C) in Clementine, Dancy and Triumph grapefruit. The presence of the different VAFs at SNP08+18bp is likely due to duplications that induce misalignments of reads during the assembly of the re-sequenced genomes by short reads technology [57]. Possible misalignments also occurred in Dancy, where a small percentage of reads with Ts at the SNP08 are present in the assembly.

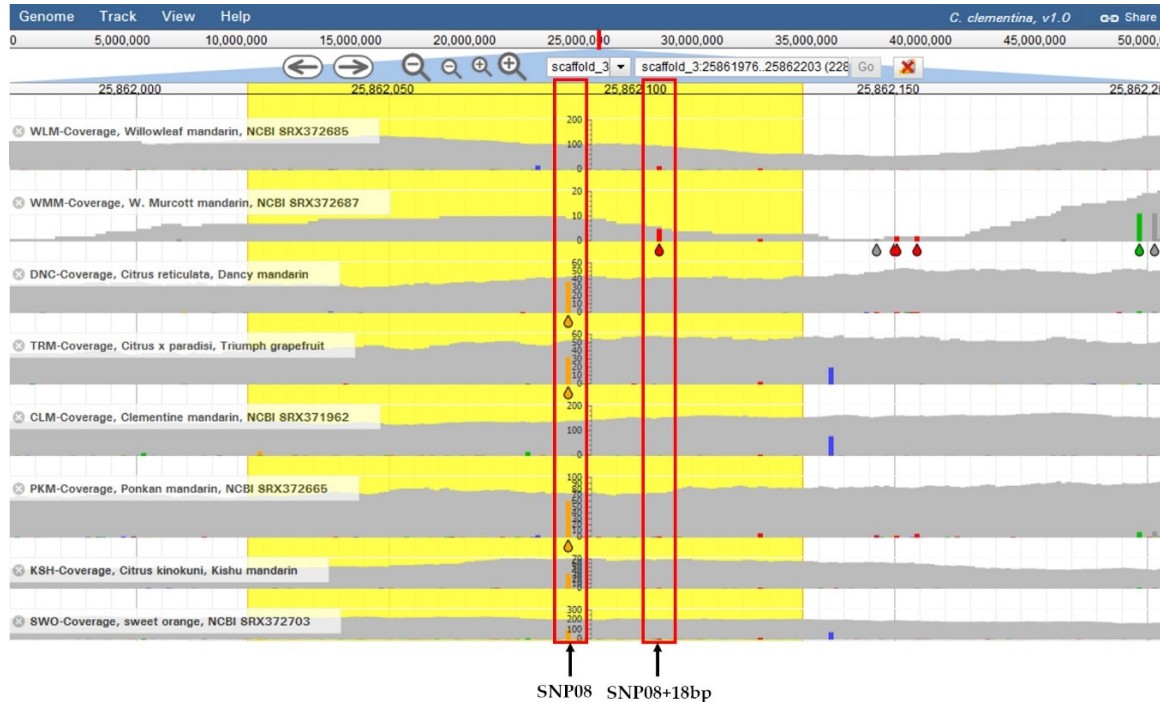

**Figure 2.** Genome browser showing the alignment of the sequence flanking the SNP08 and the additional SNP (SNP08+18bp) in resequenced genomes of Willowleaf (synonym of Avana), W. Murcott (synonym of Afourer), Dancy, Triumph grapefruit, Clementine, Ponkan, Seedless Kishu, and sweet orange aligned using the *C. clementina* genome as reference (retrieved from the citrus genome database—www.citrusgenomedb.org). The presence of different variant allele frequencies is clear at the position SNP08+18bp, particularly in Willowleaf and Ponkan mandarins, probably due to misaligned short reads not correctly assigned to other duplicated regions. Moreover, the susceptible Dancy has a small percentage of Ts at the SNP08.

To discover the duplications, BLAST searches against the genome regions flanking SNP08 in the clementine, and pummelo (*C. maxima*) genomes were performed. The analysis identified several duplications in both genomes (Figure 3). In clementine, there are four strictly associated duplications in a range of around 192 Kb of 'scaffold 3', while in pummelo, there are three duplications in a range of around 137 kb of chromosome 2. We also found duplications of 300 bp in chromosome 8 and 9 of pummelo, and in 'scaffold 8' and 'scaffold 9' of clementine, with a bit score of more than 200.

*3.2. HRM Genotyping of the Germplasm Collection*

HRM was first performed using the SNP08_HRM primers (Table 2) with a single-round protocol. However, we obtained several melt curve profiles that were difficult to interpret (data not shown). Given the presence of duplicated regions, a nested PCR protocol was used to obtain a locus-specific amplicon and optimize HRM amplification. Nested PCR coupled with HRM has been successfully applied in other studies to increase amplification specificity [58–60]. Because of the large size (3 kb) of the duplication region and the very high similarity between them, it was not possible to anchor the forward and/or reverse primers in non-duplicated regions.

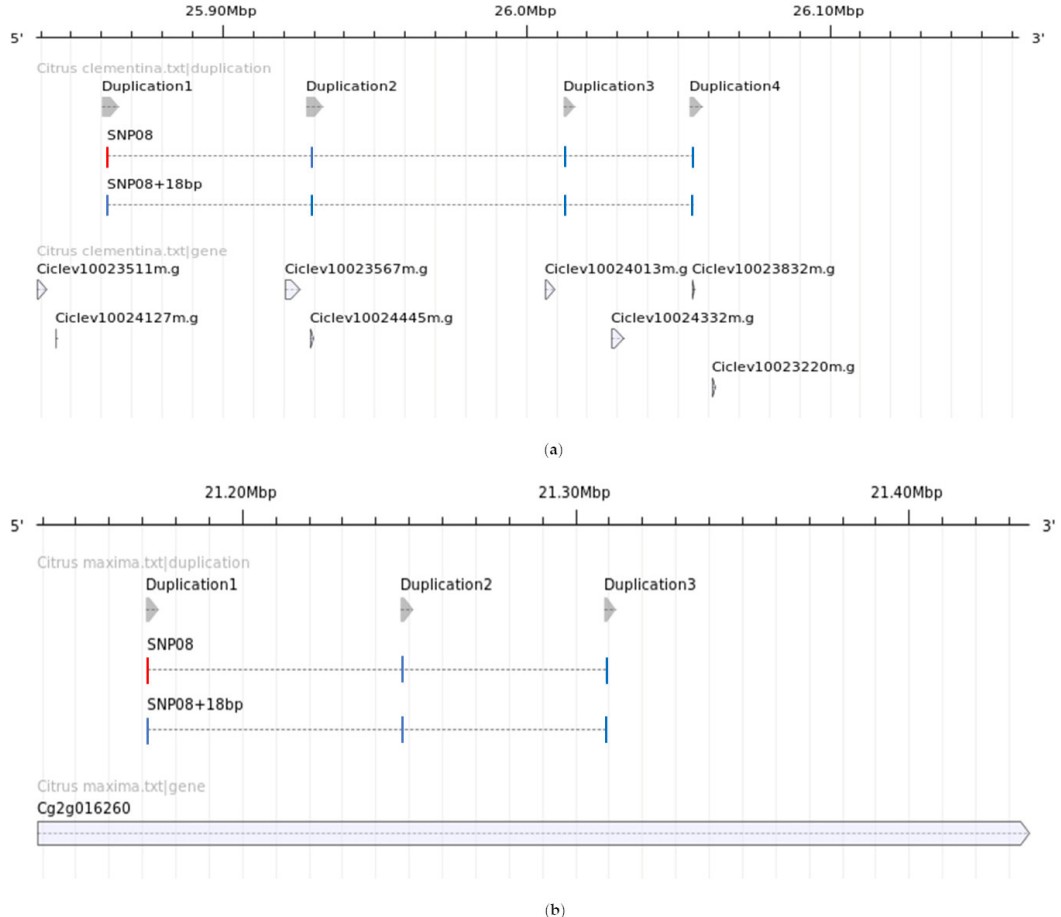

**Figure 3.** Duplicated regions flanking the SNP08 locus in *Citrus clementina* (**a**) and *C. maxima* (**b**) genomes. Duplication 1 in the clementine genome contains the mapped SNP08 at position 25862085 (in red), as indicated by Cuenca et al. [5]. (**a**) Scaffold 3 (25860390 to 25865701) in the *C. clementina* genome with the duplications of the SNP08: duplication 1 (5311 bp); duplication 2 (5306 bp); duplication 3 (3621 bp), and duplication 4 (3827 bp). The positions of SNP08 inside the duplications are: 25862085; 25929427; 26012593 and 26054623, The positions of the SNP08+18bp are: 25862103; 25929445; 26012611 and 26054641. (**b**) Chromosome 2 (21171193 to 21174566) in the *C. maxima* genome with the duplications of the SNP08: duplication 1 (3373 bp); duplication 2 (3353 bp) and duplication 3 (3209 bp). The positions of SNP08 inside the duplications are: 21171551; 21248017 and 21308902. The positions of the SNP08+18bp are: 21171569; 21248035 and 21308920. BLAST parameters used: score equal to zero and minimum homology length higher than 3000 bp.

The nested protocol was first applied to the mandarin collection, as well as to one sweet orange and one grapefruit variety (Table 1). From the analysis of the nested PCR coupled with the HRM of these 41 genotypes, six melting curve profiles were found (described in Table 3) and compared with the Sanger sequences (Figure 1) and the data available in the citrus genome database (Figure 2). The six profiles were the following: profile 1A (T/T at SNP08 + C/C at SNP08+18bp), typical of resistant accessions, like ISA, Foma107, Wilking, and Okitsu; profile 1B (T/T at SNP08 + C/T at SNP08+18bp), found in resistant varieties like Cami, Avana, King, and Fremont; profile 1C (T/T at SNP08 + T/T at SNP08+18bp) found only in the resistant Afourer; profile 2A (G/T at SNP08 + C/C at SNP08+18bp) found in Fortune, Seedless Kishu, Page, and other susceptible mandarin varieties, plus grapefruit and in sweet orange; profile 2B (G/T at SNP08 + C/T at SNP08+18bp) found in Murcott and Ponkan; profile 3A (G/G at SNP08 + C/C at SNP08+18bp), found in the susceptible Dancy, Malvasio, and Minneola.

**Table 3.** Genotypes of 41 accessions of CREA germplasm collection at the locus surrounding the SNP08, based on HRM and Sanger sequencing.

| Genotypes | SNP08 | SNP08+18bp | Genotype Code |
|---|---|---|---|
| Clemapo | TT | CC | 1A |
| Fallglo | TT | CC | 1A |
| Foma107 | TT | CC | 1A |
| ISA | TT | CC | 1A |
| Kara | TT | CC | 1A |
| Kinnow | TT | CC | 1A |
| Mapo | TT | CC | 1A |
| Okitsu | TT | CC | 1A |
| OTA9 | TT | CC | 1A |
| Rubino | TT | CC | 1A |
| Simeto | TT | CC | 1A |
| Wilking | TT | CC | 1A |
| 50-15A-6 | TT | CT | 1B |
| Avana | TT | CT | 1B |
| Cami | TT | CT | 1B |
| Carvalhais | TT | CT | 1B |
| Encore | TT | CT | 1B |
| Fremont | TT | CT | 1B |
| King | TT | CT | 1B |
| Palazzelli | TT | CT | 1B |
| Afourer | TT | TT | 1C |
| Bower | GT | CC | 2A |
| Daisy | GT | CC | 2A |
| Doppio Sanguigno | GT | CC | 2A |
| Fairchild | GT | CC | 2A |
| Fortune | GT | CC | 2A |
| Michal | GT | CC | 2A |
| Nova | GT | CC | 2A |
| Page | GT | CC | 2A |
| Primosole | GT | CC | 2A |
| Seedless Kishu | GT | CC | 2A |
| Star Ruby | GT | CC | 2A |
| Sunburst | GT | CC | 2A |
| Ellendale | GT | CT | 2B |
| Emperor | GT | CT | 2B |
| Murcott | GT | CT | 2B |
| Ortanique | GT | CT | 2B |
| Ponkan | GT | CT | 2B |
| Dancy | GG | CC | 3A |
| Malvasio | GG | CC | 3A |
| Minneola | GG | CC | 3A |

Thus, three profiles for susceptibility (two heterozygous at the SNP08 and one homozygous) and three for resistance were obtained. The six different HRM profiles can be clearly distinguished by comparison of aligned melt curves and derivative melt curves (Figure 4). The HRM profiles can clearly discriminate heterozygous individuals at the SNP08 (such as Fortune; Profile 2A) from T/T homozygous accessions (Clementine; profile 1A) and G/G homozygous accessions (Dancy; Profile 3A), because, in Fortune, the heteroduplex determines a change of the melting curve, while the two different homozygous differ for the melting temperatures. The more complex profiles with double heterozygosity, such as Murcott (Profile 2B), are also distinguishable from those with one heterozygosity (Profile 2A) for the different melt curves, as already observed in previous studies [22,29].

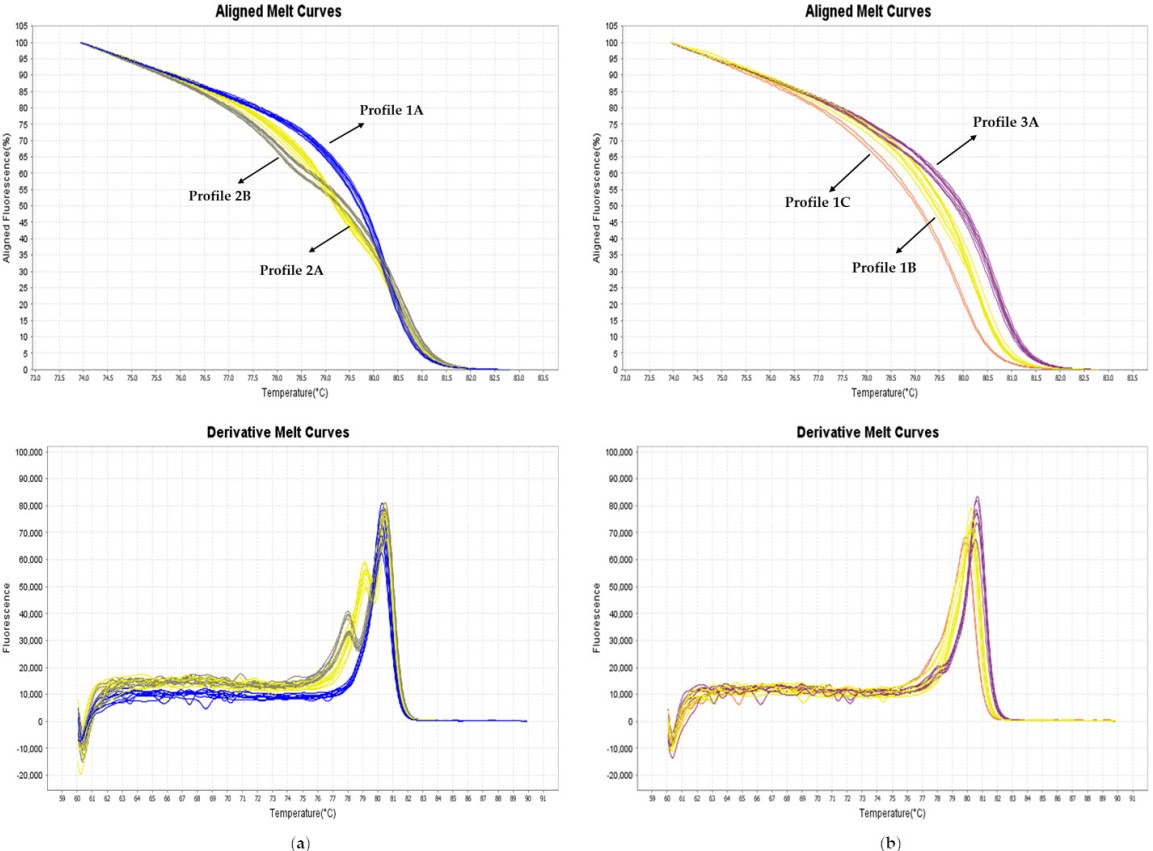

**Figure 4.** Aligned melt curves and derivative melt curves of 41 mandarin accessions of CREA germplasm collection. (**a**) Profile 1A: Clemapo, Fallglo, Foma107, ISA, Kara, Kinnow, Mapo, Okitsu, OTA9, Rubino, Simeto, Wilking; Profile 2A: Bower, Daisy, Doppio Sanguigno, Fairchild, Fortune, Michal, Nova, Page, Primosole, Seedless Kishu, Star Ruby, Sunburst; Profile 2B: Ellendale, Emperor, Murcott, Ortanique, Ponkan. (**b**) Profile 1B: 50-15-A6, Avana, Cami, Carvalhais, Encore, Fremont, King, Palazzelli; Profile 1C: Afourer; Profile 3A: Dancy, Malvasio, Minneola.

The profiles at SNP08 of the mandarin accessions matched previous results obtained by Cuenca et al. [5] using KASPar, with three exceptions. Specifically, Fremont resulted in T/T, confirming the Sanger sequencing, with the 1B profile typical of other mandarins such as Willowleaf and King, while in Cuenca et al. [5], it resulted in G/G; Fallglo resulted in T/T, with the 1A genotype, while in Cuenca et al. [5] it resulted in G/T; and Daisy resulted in G/T, with the 2A genotype, while in Cuenca et al. [5], it resulted in G/G. The HRM genotyping of these three varieties is consistent with their parentage (Table 1). The differences between the present study and the one by Cuenca et al. [5] might be due to different sources of the germplasm collection, mislabeling or cases of homonymy between the CREA and the IVIA germplasm.

Grapefruit, sweet orange, and tangors were included in the genotyping along with mandarins, because they are often used in international breeding programs to generate mandarin-like hybrids. Regarding grapefruit, the species shows symptoms both under field conditions [11] and after artificial inoculation [12]. Regarding the result obtained in sweet orange, it is known that some cultivars are susceptible to specific isolates of the pathogen in conditions of artificial inoculations [11,12,61]. However, under field conditions, sweet orange does not show symptoms of the *A. alternata* tangerine pathotype, even though ACT-producing isolates can colonize saprophytically lesions originated by abiotic disorders or pests. Pegg [8] demonstrated the role of cuticle thickness in ABS symptom expression, and Gardner et al. [62] observed a different degree of susceptibility to the fungus in young and old sweet orange leaves, suggesting that old sweet orange leaves are less permeable to the ACT than

other genotypes. Older tissues with thick cuticle permeability may play a key-role in susceptibility to the toxin, although there might be additional mechanisms for toxin inactivation. The genetic susceptibility of sweet orange to ABS is confirmed by the fact that it confers susceptibility to its progenies when crossed with resistant parents. In the framework of the IVIA breeding program [63], many triploid hybrids have been recovered in diploid by diploid and interploidy hybridizations between several clementines, as female parents with different sweet oranges as male parents. After in vitro inoculation of detached leaves and under field conditions, susceptible triploid hybrids were identified segregating as expected in hybridizations between resistant (TT) and susceptible (GT) parents, and confirming that sweet orange was the source of susceptibility (Aleza and Cuenca, personal communication).

Regarding tangors, Cuenca et al. [5] found that Ellendale tangor had the G/T genotype at the SNP08, and our results confirm this, although it is considered field resistant in all countries other than Israel [11], and produced no symptoms in the artificial inoculation studies of Vicent et al. [12]. In contrast to that observed in sweet orange, Ellendale tangor has never shown ABS symptoms during more than 100 years of commercial production under high disease pressure in Australia, and has been the main source of ABS resistance in their mandarin breeding program [16]. Despite having the G/T susceptible genotype, Ellendale performs as resistant when used as a parent. A similar case is represented by another tangor, Ortanique, which has the G/T susceptible genotype, but is field resistant.

A small number of other varieties, such as Hickson, used in the Australian breeding program, have also shown disease severity and/or breeding performance that is inconsistent with their SNP08 G/T KASPar genotyping (data not shown). This requires further breeding and molecular studies, because although the anomaly occurs in only a few varieties, some of them are useful parents for many traits and they may possess some additional mechanisms conferring disease resistance.

### 3.3. HRM and KASPar Genotyping of the Hybrid Populations

Once clear HRM profiles for the 41 accessions from the CREA germplasm collection were obtained (including many parents of the CREA breeding program, Table 1), the protocol was then applied to 862 hybrids from five populations having susceptible male or female parents heterozygous at SNP08 (Table 4).

**Table 4.** Populations and total number of hybrids genotyped at the SNP08 locus using HRM, chi-square tests and P values for each population.

| Parent Combinations | Hybrids Analysed | Resistants | % | Susceptibles | % | Chi-Square | *p* Value |
|---|---|---|---|---|---|---|---|
| ISA × Seedless Kishu | 351 | 186 | 53.0 | 165 | 47.0 | 1.256 | 0.262 |
| Foma107 × Seedless Kishu | 162 | 74 | 45.7 | 88 | 54.3 | 1.210 | 0.271 |
| Cami × Seedless Kishu | 84 | 39 | 46.4 | 45 | 53.6 | 0.429 | 0.513 |
| Rubino × Murcott | 148 | 54 | 36.5 | 94 | 63.5 | 10.811 | 0.001 |
| Fortune × Ota9 | 117 | 60 | 51.3 | 57 | 48.7 | 0.077 | 0.782 |
| Total | 862 | 413 | 47.9 | 449 | 52.1 | 1.503 | 0.220 |

Figure 5 shows the HRM curves generated in these hybrid populations. The populations of ISA × Seedless Kishu and Foma107 × Seedless Kishu show clear and well separated profiles (Figure 5a), due to the homozygosity of the SNP08+18bp in both parents. The profiles were 1A for the resistant and 2A for the susceptible hybrids. In the ISA × Seedless Kishu population, from a total of 351 hybrids analyzed, 186 (53.0%) were resistant, while 165 (47.0%) were susceptible (Table 4). For the Foma107 × Seedless Kishu population, 74 out of a total of 162 individuals were resistant (45.7%), while 88 (54.3%) were susceptible (Table 4).

The Fortune × OTA9 population also gave HRM results that were easy to interpret, with just two distinct profiles: profile 2A, typical of the susceptible parents heterozygous at the SNP08, and profile 1A in OTA9, typical of resistant accessions (Figure 5b). From a total of 117 individuals, 51.3% were resistant, while 48.7% were susceptible (Table 4).

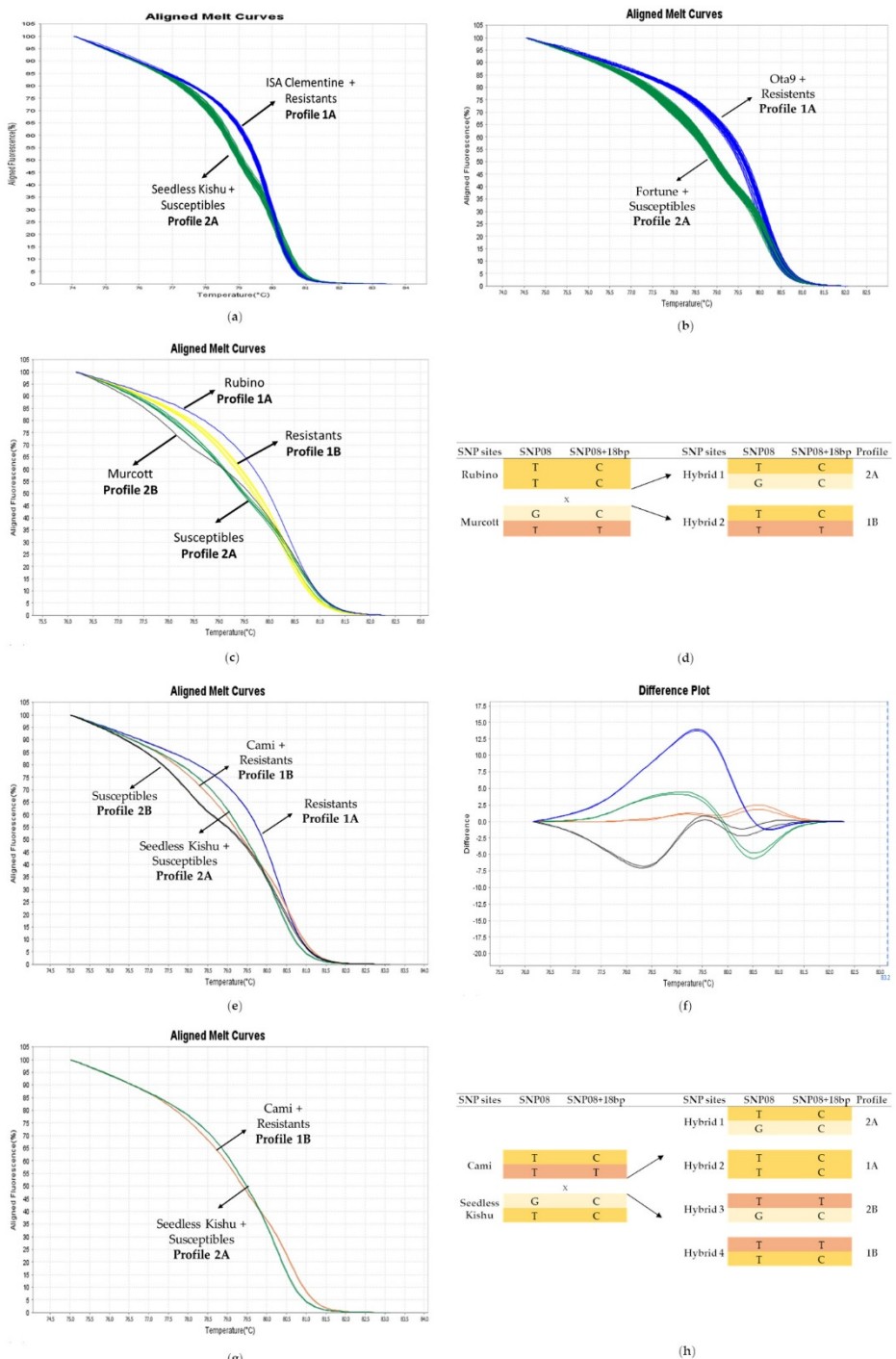

**Figure 5.** Aligned melt curves of resistant and susceptible mandarin hybrids. (**a**) ISA × Seedless Kishu; (**b**) Fortune × Ota9; (**c**) Rubino, Murcott and their hybrids; (**d**) segregation scheme of the Rubino × Murcott population based on the HRM genotyping and Sanger sequencing results; (**e**) Cami × Seedless Kishu melt curves, showing four different profiles in the progenies; (**f**) Cami × Seedless Kishu difference plot, facilitating the interpretation of the genotyping results; (**g**) melt curves of 1B and 2A hybrids of the Cami × Seedless Kishu population, showing slightly different profiles; (**h**) segregation scheme of Cami × Seedless Kishu based on the HRM genotyping and Sanger sequencing results.

Rubino × Murcott hybrids showed different curves compared to those of the parents. Specifically, both classes of hybrids displayed heterozygous profiles, as shown in Figure 5c,d: profile 1B for resistant individuals (with heterozygosity at SNP08+18bp) and profile 2A for susceptible individuals

(with heterozygosity at SNP08). The segregation profiles were different to those of the parents, so including 1B and 2A accessions as references helped in the correct interpretation of the results. It was possible to distinguish two different groups of melting curves: the first with 54 (36.5%) resistant individuals (T/T at SNP08) and the second with 94 (63.5%) susceptible individuals (G/T at SNP08) (Table 4). The interpretation of these results was more difficult than in the previous population, due to the presence of two different types of heterozygous HRM profiles.

The HRM genotyping results of the Cami (1B) × Seedless Kishu (2A) population were the most difficult to interpret, since we observed four different profiles: 1A and 1B for the resistant hybrids, as showed in 39 (46.4%) out of 84 analyzed, and 2A and 2B for the susceptible genotypes, with 45 (53.6%) out of a total of 84 hybrids analyzed (Figure 5e). The presence of multiple profiles might be due to the duplication of the SNP08 locus and the different allelic frequencies at the SNP08+18bp observed in the 2A accessions of the germplasm. As for the Rubino × Murcott population, varieties of the CREA germplasm with 1A and 2B genotypes (not found in the parents) were included in the HRM plates as additional references, to facilitate the interpretation of the results, thus making possible the detection of resistant versus susceptible F1s. Although the profiles were often influenced by the presence of the SNP08+18bp, we were able to identify the HRM curves associated with susceptible or resistant hybrids (Figure 5g). The generation of four different profiles in the F1s of this parent combination suggests the presence of additional duplications in other chromosomes (Figure 5h).

From the chi-square analysis, four out of five populations showed Mendelian segregation at the locus analyzed close to the expected 1:1 ratio, while the population involving Murcott as male parent resulted in statistically significant segregation distortion toward the susceptible hybrids ($p = 0.001$) (Table 4). To verify the reliability of HRM genotyping at the SNP08 locus, we compared the HRM results of this population with the KASPar genotyping technique (Figure 6), which was used successfully by Cuenca et al. [5], and is routinely applied in the IVIA breeding program. The results showed two separate clusters clearly distinguishing resistant (54) and susceptible (94) individuals. The data obtained with the KASPar technique matched the HRM genotyping, thus confirming that the observed segregation distortion is not due to genotyping methods. Instead, it may be due to meiotic, gametic or zygotic events [64]. Segregation distortion is frequently observed in citrus and can be caused by different mechanisms. Recently, Garavello et al. [65] indicated that it could be related to gamete selection during pollen meiotic process, or to the S-locus ribonuclease (S-RNase)-based gametophytic self-incompatibility system [66]. In this context, Rubino and Murcott both have *C. sinensis* as a parent and could share a S-RNase haplotype, thus modifying the Mendelian segregation ratio.

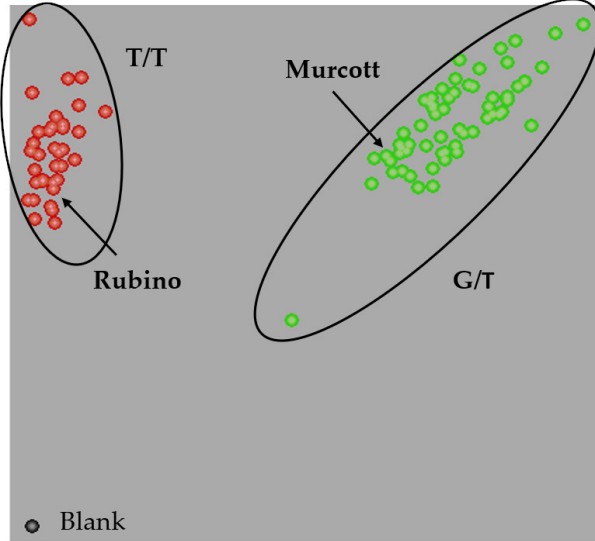

**Figure 6.** KASPar genotyping results for the SNP08 for the Rubino × Murcott population. Resistant hybrids are represented by red points (T/T); susceptible hybrids are represented by green points (G/T).

### 3.4. CAPS Assay

In order to facilitate the identification of resistant individuals and to avoid problems caused by the heterozygosity of SNP08+18bp (as particularly observed in the Cami × Seedless Kishu population), a CAPS marker (SNP08-CAPS) that could be used as an alternative to HRM was developed. The SNP08-CAPS was developed, starting from the nested PCR product (572 bp). From the virtual digestion in Benchling, two restriction sites were identified for the susceptible allele, GATGG the binding site of *Bcc*I and GCGATG the binding site of *Btg*ZI (Figure 7). *Bcc*I was chosen because it is less costly. The *Bcc*I enzyme recognizes the region of five nucleotides from position 271 to 275 of the amplicon, which corresponds to the position of the SNP08 in the presence of G of the susceptible allele, and cleaves it in position 266 of the amplicon, producing two fragments of 265 bp and 307 bp. Three results are therefore expected: (1) a non-digested single band in the case of T/T homozygosity, (2) three bands from heterozygous susceptible genotypes (G/T), and (3) two bands resulting from complete digestion in the G/G genotypes.

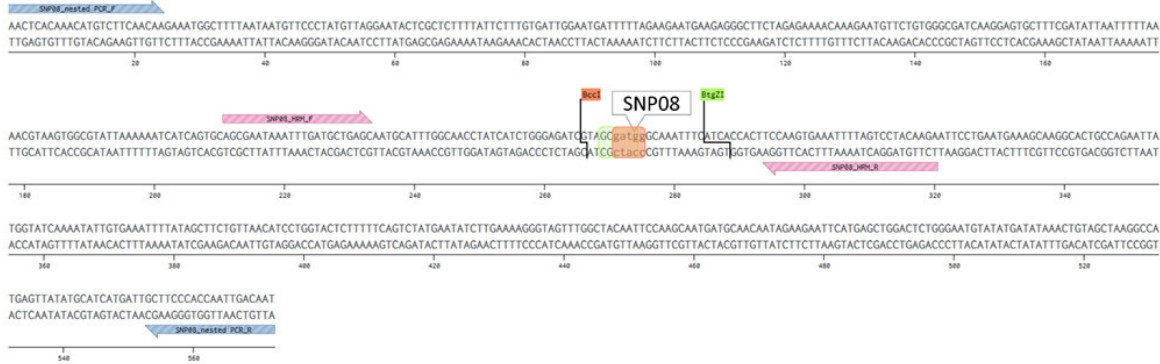

**Figure 7.** Amplicon with related primers; in blue, the first cycle of nested PCR, and fuchsia, the second cycle, respectively, carried out in a standard and HRM thermocycler. The *Bcc*I and *Btg*ZI binding and cutting sites, respectively, in dark pink and light green.

The band patterns obtained are shown in Figure 8. While the pattern was as expected in the T/T and the G/T accessions, the G/G showed the same pattern of G/T (Figure 8a). The presence of the undigested band is unlikely to be the result of partial digestions, because the digestion protocol was modified by increasing the incubation step and changing the enzyme units, without having any effect on the banding patterns (data not shown). Therefore, the undigested bands are likely to indicate that the allele with T is present in some of the duplications. Sequence read archive (SRA) data of Dancy showed in Figure 2 seems to confirm our hypothesis, where a small proportion of Ts could be observed at the SNP08 position.

Despite the limitation in discriminating G/T from G/G accessions, the SNP08-CAPS has the advantage of distinguishing between resistant and susceptible individuals based on just two banding patterns. This has important practical significance in breeding programs where the objective is simply to identify and retain only those hybrids that are resistant. Figure 8b shows the banding patterns obtained in the Rubino × Murcott population, where, unlike the HRM profile, resistant hybrids have the same profile as Rubino, and susceptible ones have the same profile as the male parent Murcott. A similar result was obtained in the Cami × Seedless Kishu hybrids (Figure 8c), in which the use of SNP08-CAPS greatly facilitated the interpretation compared to HRM.

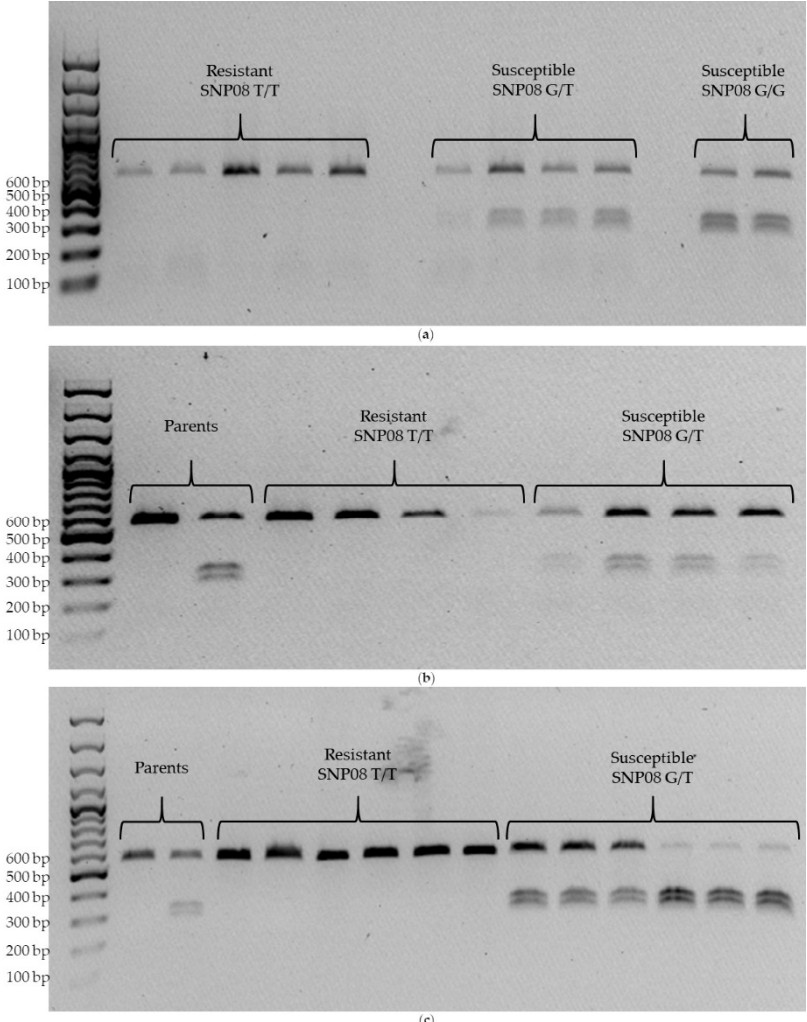

**Figure 8.** Digestion with *BccI* enzyme. (**a**) Profile of resistant and susceptible accessions of the CREA germplasm. In order: lane 1, 100 bp marker (GeneRuler DNA Ladders, ThermoFisher); lane 2, Afourer (Profile 1C in HRM); lane 3 and 4, ISA and Foma107 (Profile 1A in HRM); lane 5 and 6, Cami and King (Profile 1B in HRM); lane 8 and 9, Seedless Kishu and Fortune (Profile 2A in HRM); lane 10 and 11, Murcott and Ponkan (Profile 2B in HRM); lane 13 and 14, Dancy and Minneola (Profile 3A in HRM). (**b**) Profile of resistant and susceptible hybrids of Rubino × Murcott. In order: lane 1, 100 bp marker (GeneRuler DNA Ladders, ThermoFisher); lane 2, Rubino; lane 3, Murcott, lane 4 to lane 7, resistant hybrids; lane 8 to 11, susceptible hybrids. (**c**) Profile of resistant and susceptible hybrids of Cami × Seedless Kishu. In order: lane 1, 100 bp marker (GeneRuler DNA Ladders, ThermoFisher); lane 2, Cami; lane 3, Seedless Kishu, lane 4 to lane 9, resistant hybrids; lane 10 to 15, susceptible hybrids.

### 3.5. In Vitro Phenotyping

In vitro phenotyping tests were performed to verify the reliability of the genotyping methods. Young leaves of 101 hybrids, randomly selected from susceptible and resistant F1s from the five populations, were artificially inoculated with *A. alternata* spores ($1 \times 10^5$ conidia-mL$^{-1}$), and the disease symptoms were evaluated after 2 to 4 days (Figure 9). Resistant and susceptible individuals were found in all F1s tested (Table 5). In addition, it was observed that the degree of susceptibility to *A. alternata* spores was different among populations, with the Foma107 × Seedless Kishu population giving a severe necrotic reaction on many hybrids (Figure 9i). Symptoms were less severe and developed more slowly on other hybrids (Figure 9a,c,e,g), making it necessary to continue evaluations for 4 days in order to avoid disease-escapes. Spore inoculation has been widely used in ABS research for more than

50 years [5,8,11,12,16], and is a reliable method of identifying susceptible genotypes, even those that do not normally develop symptoms under field conditions. In our tests, necrotic symptoms developed on leaves from all 48 hybrids with the G/T genotype at SNP08, while leaves from the 53 hybrids with the T/T genotype remained healthy. The results obtained from the analyzed subset were totally in agreement with the genotyping results at the SNP08 locus, confirming the usefulness of the marker for the selection of resistant hybrids in these families.

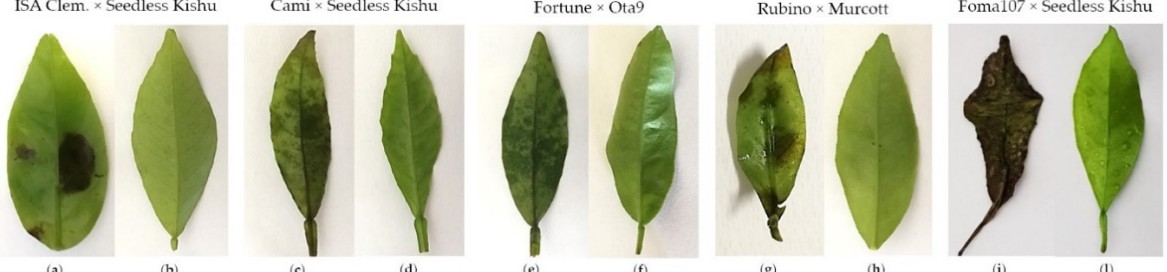

**Figure 9.** Leaves of resistant and susceptible mandarin hybrids showing a range of necrosis severity four days after inoculation with *A. alternata* spores. The leaves belong to hybrids of ISA × Seedless Kishu (**a,b**), Cami × Seedless Kishu (**c,d**), Fortune × OTA9 (**e,f**), Rubino × Murcott (**g,h**), and Foma107 × Seedless Kishu (**i,l**) populations. a, c, e, g, i: susceptible hybrids; b, d, f, h, l: resistant hybrids.

**Table 5.** In vitro phenotyping of hybrid populations for Alternaria brown spot resistance.

| Parent Combinations | Hybrids Analysed | Resistants | % | Susceptibles | % | Chi-Square | *p* Value |
|---|---|---|---|---|---|---|---|
| ISA × Seedless Kishu | 14 | 10 | 71.4 | 4 | 28.6 | 2.571 | 0.109 |
| Foma107 × Seedless Kishu | 20 | 10 | 50.0 | 10 | 50.0 | 0.000 | 1.000 |
| Cami × Seedless Kishu | 20 | 12 | 60.0 | 8 | 40.0 | 0.800 | 0.371 |
| Rubino × Murcott | 26 | 8 | 30.8 | 18 | 69.2 | 3.846 | 0.050 |
| Fortune × Ota9 | 21 | 13 | 61.9 | 8 | 38.1 | 1.190 | 0.275 |
| Total | 101 | 53 | 52.5 | 48 | 47.5 | 0.248 | 0.619 |

## 4. Conclusions

We confirmed that SNP08 is a useful and reliable tool to discriminate resistant and susceptible progenies. However, it is surrounded by highly duplicated regions, which complicate the interpretation of single-round HRM profiles. Moreover, we found an additional SNP 18bp downstream of SNP08 (called SNP08+18bp), that is not associated with resistance/susceptibility, but hampered the analysis of SNP08 with HRM, and in some cases, resulted in complex heterozygous genotypes in the F1s. Despite this complexity of the genome region surrounding SNP08 and the presence of an additional SNP close to SNP08, we were able to optimize a protocol based on PCR and HRM for reliable disease screening of different mandarin populations from diverse genetic backgrounds. The nested protocol reduced the complexity of the amplified region and allowed the generation of clear HRM profiles. The HRM genotyping was validated using KASPar results and by leaf phenotyping, thus confirming the reliability of the methods. However, when using specific parents with the 2A genotype, the detection of susceptible versus resistant hybrids was possible (especially including additional controls in each plate), but more complicated, due to the generation of slightly different heterozygous melting curves. In this case, the use of CAPS overcomes the problem of complex HRM profiles, generating only two clear banding patterns, one for the resistant and one for the susceptible hybrids. Thus, both protocols should prove useful in disease resistance breeding programs, with CAPS allowing the simple culling of all susceptible hybrids, while HRM provides additional information that may sometimes be needed for parental selection. Based on the above considerations, the choice of HRM or CAPS protocols will depend on the genotype of the parents at the SNP08 locus, the purpose of screening, and the resources and facilities available for marker assisted selection.

**Author Contributions:** Investigation, formal analysis, writing—original draft preparation, C.A.; data curation, formal analysis, methodology, A.C.; investigation, methodology, writing—original draft preparation, M.C.S.; investigation, V.C.; investigation, F.S.; writing—review and editing, P.C.; writing—review and editing, resources, C.L.; funding acquisition, resources, G.R.; Validation, writing—review and editing, M.W.S.; validation, J.C.; validation, writing—review and editing, P.A.; Conceptualization, methodology, writing—original draft preparation, resources, supervision, M.C. All authors have read and agreed to the published version of the manuscript.

**Funding:** This research was funded by "Convenzione CREA-Fruitimpresa" and "Convenzione CREA-Armonia".

**Acknowledgments:** The authors would like to thank Gaetano Distefano for the technical support in the HRM analysis.

**Conflicts of Interest:** The authors declare no conflict of interest.

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
