# Peer review of "Disease Resistant Citrus Breeding Using Newly Developed High Resolution Melting and CAPS Protocols for Alternaria Brown Spot Marker Assisted Selection"

_agronomy, doi:10.3390/agronomy10091368_

Round 1

Reviewer 1 Report

In this manuscript the authors have evaluated disease resistant citrus breeding using newly developed High Resolution Melting and CAPS protocols for Alternaria brown spot marker assisted selection. My recommendation is reconsider after major revision and below please find my comments and suggestions.

The authors should use the help of a native English speaker for proofreading. For the introduction and material and methods I only have a few comments to be addressed and they are listed in the pdf document.

In the results and discussion, the authors need to discuss better their results and make a better job in their story telling. It seems like they are most of the times only describing the results and not discussing the reason behind their findings. Please include more discussion in this section.

Line 241: the authors described a 18bp region downstream SNP8. Can you explain it in more details? Why were you looking at this region? How was it detected? Are there any other region flanking SNP8 which contain other SNPs?

Line 257: Heterozygosity based on misalignments; how can it be fixed? If it is not misalignment what might be going on?

Line 468: In your CAPS gels you indicate that the undigested bands in Fig 8a is not due to partial digestion. However, it seems like you dont have a positive control showing that the enzyme was capable of totally digesting a DNA. Can you include that here?

Line 510: Are you sure this experiment has worked? I dont see any typical sympotms of Alternaria there. How can you prove that is has worked?

In conclusion, the information contained in the manuscript would be very useful for the scientific community, however, it needs a bit of improvement before can be recommended for publication.

Author Response

In this manuscript the authors have evaluated disease resistant citrus breeding using newly developed High Resolution Melting and CAPS protocols for Alternaria brown spot marker assisted selection. My recommendation is reconsider after major revision and below please find my comments and suggestions.

The authors should use the help of a native English speaker for proofreading. For the introduction and material and methods I only have a few comments to be addressed and they are listed in the pdf document.

We revised the language carefully throughout the manuscript. One native English speaker is among the authors of the manuscript.

In the results and discussion, the authors need to discuss better their results and make a better job in their story telling. It seems like they are most of the times only describing the results and not discussing the reason behind their findings. Please include more discussion in this section.

We made changes in the results and discussion as suggested in specific parts (see below for the detailed changes) and welcome the opportunity to expand our explanation of the results. We would like to draw your attention to significant discussion of the results on aspects such as:

  1. the structural analysis of the SNP08 locus,
  2. the reliability of the SNP08 marker,
  3. discrepancies between SNP genotyping and ABS resistance,
  4. the segregation distortion observed in one population,
  5. the leaf inoculation protocol, and
  6. the advantages and disadvantages of the two proposed methods.

All of these elements are woven into a story about how the new methods can be effectively applied to disease resistance breeding in citrus, using marker assisted selection.

Line 241: the authors described a 18bp region downstream SNP8. Can you explain it in more details? Why were you looking at this region? How was it detected? Are there any other region flanking SNP8 which contain other SNPs?

We have inserted additional discussion to better explain the detection and significance of the SNP at 18bp downstream of SNP08.  We were not looking for the additional SNP, but we found one while sequencing the region flanking SNP08.  Sanger sequencing is a prerequisite of HRM analysis because without knowing all the SNP sites and their possible allelic combinations, it is not possible to correctly interpret the HRM curves. Other than SNP08 and SNP08+18bp, there are no additional SNPs in the region amplified with the HRM forward and reverse primers. Mandarins are highly heterozygous, with sometimes as many as 17 heterozygous SNP sites per Kb (Wu et al 2014, Nat Genet), so it is common to find more than one SNP in regions of a few hundred base pairs.

Line 257: Heterozygosity based on misalignments; how can it be fixed? If it is not misalignment what might be going on?

We rephrased the sentence (lines 271-273) and added a reference:  “The presence of the different VAFs at SNP08+18bp is likely due to duplications that induce misalignments of reads during the assembly of the re-sequenced genomes by short reads technology (Treangen 2012).”. Misalignments might happen in duplicated regions when re-sequenced genomes are based on the alignments of (Illumina) short reads.

Line 468: In your CAPS gels you indicate that the undigested bands in Fig 8a is not due to partial digestion. However, it seems like you don’t have a positive control showing that the enzyme was capable of totally digesting a DNA. Can you include that here?

We modified the sentence in the manuscript, to better explain the significance of the undigested bands (lines 485-492). We do not have any samples with totally digested bands. In the GG samples at the SNP08, there are likely duplications in which TT or TG allelic configurations are present, so the undigested band is always present. Duplications are the obstacle to obtaining three allelic profiles for TT, GT and GG accessions and this is a recognised limitation of the CAPS method (as we state in the manuscript).  However, we also indicate that the CAPS marker is useful to discriminate resistant (TT) from the susceptible (GT or GG) hybrids, which is often the primary purpose of marker assisted selection.  When it is necessary to discriminate GG from GT genotypes (such as in parent selection), then HRM is the method to choose.

Line 510: Are you sure this experiment has worked? I don’t see any typical symptoms of Alternaria there. How can you prove that is has worked?

The symptoms visible after in vitro leaf inoculation, for the peculiar conditions of the test (optimal humidity and temperature, concentration of conidia, stage of leaf growth), cannot be compared to the usual symptoms that occur in the field. We believe that our method is reliable, since similar tests have been used worldwide for more than 50 years. When we use in vitro screening for ABS resistance, we often get symptoms that are different from field symptoms, but there is no doubt that these necrotic symptoms are indicative of a susceptible genotypes. We rephrased the sentence to clarify the concerns of the reviewer (section 3.6, lines 522-529):

“In addition, it was observed that the degree of susceptibility to A. alternata spores was different among populations, with Rubino x Murcott population giving a severe necrotic reaction on many hybrids (Figure 9j). Symptoms were less severe and developed more slowly on other hybrids (Figure 9a-c-e-g) making it necessary to continue evaluations for 4 days in order to avoid disease-escapes. Spore inoculation has been widely used in ABS research for more than 50 years (Pegg 1966, Solel 1997, Vicent 2004, Miles 2015, Cuenca 2016) and is a reliable method of identifying susceptible genotypes, even those that do not normally develop symptoms under field conditions. In our tests, necrotic symptoms developed on leaves from all 48 hybrids with the G/T genotype at SNP08, while leaves from the 53 hybrids with the T/T genotype remained healthy.”

We also changed the caption of Figure 9, and replaced some photos of figure 9, including some that better represent the symptoms obtained.

In conclusion, the information contained in the manuscript would be very useful for the scientific community, however, it needs a bit of improvement before can be recommended for publication.

We are grateful to Reviewer 1 that considered our manuscript very useful for the scientific community. We thank him for his suggestions that helped to improve the quality of our manuscript.

Other comments indicated in the pdf file:

Line 24: We indicated the full name of the acronym ACT (Alternaria citri toxin)

Line 40: We indicated the full names of the acronym KASPar and SNP

Line 54: We indicated the full name of GWAS

Lines 73-74: We clarified why MAS for ABS resistance might help sustain modern citriculture. We completed the sentence as follow: “Performing MAS for ABS resistance might help sustain modern citriculture facilitating the generation of new resistant citrus cultivars.”

Line 77-78: We replaced “might” with “is”. We added a reference about the usefulness of MAS in citrus breeding.

Lines 82-86: We included two sentences that concisely explaining the KASPar technique.

Line 94. We already provide many references for HRM and CAPS when specifically discussing each technique in the following two paragraphs. Adding references about HRM and CAPS at this point makes the paragraph confusing.

Line 123: We added punctuation as requested.

Table 1: “Possibly” means that Afourer is a chance seedling of Murcott, with Mandalina reported as the possible male parent.   

Table 1: We deleted the taxonomic information about the analysed accessions as requested.

Line 145: We changed “rpm” to “g”.

Line 192: We added the URL of the software as requested.

Line 253-254: We changed the sentence as requested.

Line 256: See the explanation above.

Lines 259, 266, 270, 272, 435: Dosage is a term that is not restricted to gene expression. The term ‘allele dosage’ is used in the bibliography to indicate the copy number of each allele. The term is commonly used in cases of gene duplications or genome duplications, particularly when heterozygosity differs from the expected 1:1 ratio (for example, see McKinney, G. J., Waples, R. K., Pascal, C. E., Seeb, L. W., & Seeb, J. E. (2018). Resolving allele dosage in duplicated loci using genotyping‐by‐sequencing data: A path forward for population genetic analysis. Molecular Ecology Resources, 18, 570– 579).

However, following the suggestion of the reviewer and to simplify the manuscript for all readers, we changed the term “dosage”. Consequently, when we refer to Sanger sequencing we use the term “frequencies” instead of “dosages”, line 259); when we refer to resequencing data, we use “different variant allele frequencies (VAFs)” lines 266 to 273) and; when we talk about the results of  HRM profiles, at line 435, we replaced “dosages” with “different allelic frequencies at the SNP08+18bp”.

Line 259: We deleted “while”.

Line 271: We replaced “while” with “and”.

Line 271-273: See the explanation above.

Line 280: we added Ponkan mandarin.

Line 309: We changed “not easy” to “difficult”.

Lines 352-360: We rephrased the paragraph specifying the differences between the present study and the previous one by Cuenca and colleagues.

Line 361-388: We made modifications throughout the whole paragraph to make it clearer and more connected with the previous discussion.

Line 372: We changed the sentence to “although there might be additional mechanisms for toxin inactivation”

Line 390: We changed “disease reaction” to “disease severity”

Line 418: We changed “that could be easily interpreted” to “that were easy to interpret”.

Line 429: “was” refers to “interpretation”.

Line 454: We added “S-locus ribonuclease”.

Line 471: We replaced “ for its lower price” with “because it is less costly”.

Section 3.6: See explanation above (lines 522-529)

Reviewer 2 Report

Dear author,

Kindly see attached the document with my comments.

Best regards

Reviewer 3 Report

Overall, the paper is well structured and full of details, especially as regards the methodology and description of the results. It is only suggested to add a brief mention of the economic importance of the productions that can be damaged by the pathogen examined.

I suggest adding in the introduction a brief, simple reference to the production of that species and its economic value in the countries most involved.

Author Response

We are grateful to reviewer 3 for considering our manuscript well structured and full of details.

Not all citrus species, and not all mandarin cultivars are susceptible to the pathogen. Therefore, it is not easy to find specific information about the economic importance of the susceptible cultivars.

As examples of the impact of ABS on citrus industry, we added new sentences and four references from line 64 to line 72 reporting the drastic reduction in the production of two susceptible cultivars in two countries (Emperor production in Australia and Fortune mandarin in Spain) as well as the economic cost per hectare for Murcott.

Round 2

Reviewer 1 Report

Dear authors,

Thanks for addressing my comments and concerns. I recommend it for publication.